# Endothelial Dysfunction, Inflammation and Coronary Artery Disease: Potential Biomarkers and Promising Therapeutical Approaches

**DOI:** 10.3390/ijms22083850

**Published:** 2021-04-08

**Authors:** Diana Jhoseline Medina-Leyte, Oscar Zepeda-García, Mayra Domínguez-Pérez, Antonia González-Garrido, Teresa Villarreal-Molina, Leonor Jacobo-Albavera

**Affiliations:** 1Genomics of Cardiovascular Diseases Laboratory, National Institute of Genomic Medicine (INMEGEN), Mexico City 14610, Mexico; dianajhos18@gmail.com (D.J.M.-L.); oscar.zega@gmail.com (O.Z.-G.); mdominguez@inmegen.gob.mx (M.D.-P.); gonzalezg.antonia@gmail.com (A.G.-G.); mvillareal@inmegen.gob.mx (T.V.-M.); 2Posgrado en Ciencias Biológicas, Universidad Nacional Autónoma de México (UNAM), Coyoacán, Mexico City 04510, Mexico

**Keywords:** coronary artery disease (CAD), atherosclerosis, endothelial cells (EC), endothelium, inflammation, endothelial dysfunction, novel biomarkers, therapeutics

## Abstract

Coronary artery disease (CAD) and its complications are the leading cause of death worldwide. Inflammatory activation and dysfunction of the endothelium are key events in the development and pathophysiology of atherosclerosis and are associated with an elevated risk of cardiovascular events. There is great interest to further understand the pathophysiologic mechanisms underlying endothelial dysfunction and atherosclerosis progression, and to identify novel biomarkers and therapeutic strategies to prevent endothelial dysfunction, atherosclerosis and to reduce the risk of developing CAD and its complications. The use of liquid biopsies and new molecular biology techniques have allowed the identification of a growing list of molecular and cellular markers of endothelial dysfunction, which have provided insight on the molecular basis of atherosclerosis and are potential biomarkers and therapeutic targets for the prevention and or treatment of atherosclerosis and CAD. This review describes recent information on normal vascular endothelium function, as well as traditional and novel potential biomarkers of endothelial dysfunction and inflammation, and pharmacological and non-pharmacological therapeutic strategies aimed to protect the endothelium or reverse endothelial damage, as a preventive treatment for CAD and related complications.

## 1. Introduction

Cardiovascular disease is the leading cause of death worldwide [1]. Coronary artery disease (CAD) is the most common and is characterized by the accumulation of lipids and immune cells in the subendothelial space of the coronary arteries or atherosclerosis. This process involves the inflammatory response of the vascular endothelium [2,3,4]. Endothelial cells (EC) form a semipermeable monolayer that separates the wall of the arteries from the components of intravascular flow [3,4,5]. This barrier regulates vascular tone, prevents platelet aggregation, and maintains fluid homeostasis. The endothelium produces vasodilator and vasoconstrictor molecules such as nitric oxide (NO) and endothelin, respectively; the imbalance in production of these vasoactive substances results in the loss of its function, which is defined as endothelial dysfunction [3,4,6,7]. Endothelial dysfunction plays an essential role in the development of atherosclerosis and can be triggered and exacerbated by different cardiovascular and cardiometabolic risk factors [8,9]. Currently, there is a wealth of data on endothelial dysfunction and the risk of developing atherosclerosis and CAD. The aim of this review is to describe the most important and up-to-date information on the endothelial function, potential biomarkers of endothelial dysfunction and new pharmacological and non-pharmacological promising therapeutic approaches to prevent it.

## 2. The Endothelium

The endothelium is formed by a single layer of EC located in the intima layer of the arteries (Figure 1) [10,11]. A human adult has approximately 10 billion EC which constitute about 1.5% of total body mass [12]. For many years, the endothelium was considered a simple barrier delimiting the vessel wall [10,13], but is currently known to play an important role in cardiovascular physiology and pathophysiology by regulating vascular tone, coagulation, the exchange of fluids and solutes, inflammation, and angiogenesis, among others [14,15,16,17,18,19,20,21,22,23,24,25,26].

### 2.1. Function as a Selective Barrier

The endothelium is a semi-permeable barrier that allows the passage of water and molecules smaller than 6 nm to the subendothelial space [11,30,31]. The barrier function is maintained by the glycocalyx and by protein binding complexes (Figure 2). The glycocalyx covers the luminal surface of the endothelium and contributes to vascular permeability regulation and forms a barrier against pathogens [32,33]. It is formed by negatively charged glycoproteins, proteoglycans, and glycosaminoglycans [32,34,35,36]. These negative charges repel cells such as platelets, erythrocytes, and leukocytes. On the other hand, protein binding complexes (tight junctions, adherens junctions and gap junctions) contribute to the barrier function by maintaining the continuity of the endothelium [25,31,37,38,39].

The transport of lipoproteins through the barrier and into the subendothelial space plays a pivotal role in the pathogenesis of atherosclerosis. Elevated circulating low-density lipoprotein (LDL) levels represent one of the best-characterized risk factors for CAD and the accumulation of LDL under the arterial endothelium represents the first step in the pathogenesis of atherosclerosis [41,42,43,44]. Animal models have shown that this transport occurs by transcytosis mediated in part by the SR-BI receptor, and transcytosis was found to be saturable and dynamin-dependent in human coronary cells [45]. Notably, apolipoprotein A1 (ApoA1) can also be transported to the subendothelial space by transcytosis, modulated by ATP-binding cassette transporter A1 (ABCA1) [46,47]. ApoA1 is the main protein component of high-density lipoproteins, and although its atheroprotective effect is attributed to its role in reverse cholesterol transport, other atheroprotective mechanisms of ABCA1 expression in EC have been described [48,49,50].

### 2.2. Regulation of Haemostasis and Thrombosis

Under physiological conditions, EC prevents thrombosis through different anticoagulant and antiplatelet mechanisms. One essential way in which endothelial cells regulate the clotting mechanism is by controlling the expression of binding sites for anticoagulant and procoagulant factors on the cell surface [51]. EC prevent thrombosis by providing tissue factor and thrombin inhibitors and receptors for protein C activation [52].

Over the last decade, additional genes, signaling pathways and molecules involved in the prevention of prothrombotic states have been identified [53,54,55,56]. The mitochondrial thioredoxin system prevents the formation of reactive oxygen species (ROS) in EC. The loss of thioredoxin reductase 2 (*Txnrd2*) in murine EC generated a prothrombotic endothelium and *Txnrd2* knockout mice developed microthrombi in arteries, arterioles and capillaries, suggesting that the prothrombotic phenotype is systemic [53]. In another study, endothelial-specific deletion of *ATG7* (autophagy related 7) in mice attenuated thrombosis and decreased tissue factor expression. Endothelial autophagy is a novel target for reducing arterial and venous thrombosis [55]. The Sirt1/FoxO1 pathway plays an important role in regulating endothelial cell autophagy. Using a human umbilical vein endothelial cell (HUVEC) model, Wu et al. identified that inhibition of the Sirt1/FoxO1 signaling pathway, increased the release of the pro-thrombotic von Willebrand factor, suggesting that Sirt1/FoxO1 are possible therapeutic targets for prevention of thrombosis [56]. In addition, miR-181b and Card10 (caspase recruitment domain family member 10) were found to be important regulators of thrombin-induced EC activation and arterial thrombosis, as miR-181b overexpression inhibited thrombin-induced activation of NF-kB signaling by targeting the Card10 [54].

The ongoing identification of novel genes and pathways involved in vascular endothelial function is promising for the development of new pharmacologic approaches designed to modulate endothelial activity for the treatment of cardiovascular and thrombotic disorders.

### 2.3. Regulation of Vascular Tone

In resistance arteries, the endothelium plays a fundamental role in the regulation of vascular tone, local blood flow and systemic blood pressure via the generation of various vasoactive stimuli [57]. This monolayer operates to sense, integrate and transduce signals present in the blood and local tissue environment, which then initiate dynamic modulation of contractile activity of the surrounding vascular smooth muscle. In response to mechanical (e.g., shear stress due to blood flow) and chemical (e.g., acetylcholine, bradykinin, ATP) stimuli, EC release vasodilatory factors that regulate the vascular tone [58]. The main vasoconstrictors produced by the endothelium are thromboxane A2 (TXA2) and endothelin-1 (ET-1), while the main endothelial vasodilator factors are NO, prostacyclin (PGI2), and endothelium-derived hyperpolarization factor (EDHF) [28].

Vasodilation and vasoconstriction are regulated by changes in endothelial cytoplasmic Ca^2+^ concentrations, which are in turn modulated by several signaling pathways and mechanisms. The activation of endothelial TRPV4 (transient receptor potential cation channel, member 4) channels induces vasodilation through Ca^2+^ influx and activation of inositol 1, 4, 5-triphosphate receptors that release Ca^2+^ from the endoplasmic reticulum [59]. Moreover, shear stress is an important stimulus for release of endothelium vasoactive factors such as NO, by activating the PIEZO1 (Piezo-type mechanosensitive ion channel component 1) channel, which results in the release of adrenomedullin and the downstream production of NO [60]. The disequilibrium in the production of these substances triggers endothelial dysfunction.

By regulating vascular tone, the endothelium plays an essential role in long-term cardiac performance and adequate endothelial function prevents cardiovascular events. Continued development and refinement of therapeutic strategies to prevent endothelial dysfunction or damage should thus have direct health-related benefits.

### 2.4. Endothelial Dysfunction

Endothelial dysfunction is generated when there is an imbalance in the production or bioavailability of endothelium-derived NO, generating a decreased vasodilator response and a prothrombotic and proinflammatory endothelium. During the inflammatory process induced by different risk factors as hypertension, oxidized LDL (oxLDL) and diabetes, there is an increase in the production of interleukin-1 (IL-1), interleukin-6 (IL-6), TNF-α and C-reactive protein (CRP) that generate the endothelial proinflammatory phenotype characterized by an increase in E-selectin, vascular cell adhesion molecule-1 (VCAM-1) and intercellular adhesion molecule 1 (ICAM-1) expression (Figure 3) [61,62]. Therefore, there is a greater interest in the search for new biomarkers and therapeutic strategies that help to prevent endothelial dysfunction and reduce the risk of developing CAD and its complications.

The role of ROS and increased oxidative stress is essential in endothelial dysfunction. ROS are reactive intermediates of molecular oxygen that act as important second messengers within cells; however, an imbalance between generation of reactive ROS and antioxidant defense systems represents the primary cause of endothelial dysfunction, leading to vascular damage in both metabolic and atherosclerotic diseases. In endothelial cells, NO is essential for vascular homeostasis. Reduction in NO bioavailability, resulting from reduced NO production and/or increased NO degradation by superoxide anion, marks the onset of endothelial dysfunction [65]. Identification of new endothelial dysfunction-related oxidative stress markers represents a research goal for better prevention and therapy of CVD. New-generation therapeutic approaches based on carriers, gene therapy, cardiolipin stabilizer, and enzyme inhibitors have proved useful in clinical practice to counteract endothelial dysfunction. Experimental studies are in continuous development to discover new personalized treatments [66]. Both endothelial dysfunction-related oxidative stress markers and therapeutic approaches have been thoroughly described elsewhere in extensive reviews [4,65,66,67,68,69,70,71,72,73,74].

## 3. Biomarkers

Research on biomarkers and their clinical application has grown exponentially over the last decades [75]. Systemic biomarkers have provided invaluable insight into the pathophysiology of atherosclerosis and the development of novel therapies [76]. Endothelial dysfunction and inflammation play a central role in the development and progression in CAD [77,78]. Mediators of inflammation are secreted by inflammatory and vascular cells in the atherosclerotic plaque, or by organs such as the liver or adipose tissue [79]. Thus, several inflammation-related factors are considered biomarkers for early prediction of CAD.

### 3.1. Traditional Biomarkers

Acute-phase proteins, cytokines, adhesion molecules and microparticles have been extensively studied as endothelial dysfunction and inflammation markers in clinical studies.

#### 3.1.1. Acute-Phase Proteins

*C-Reactive Protein (CRP)*. CRP is a systemic inflammatory mediator and a major acute phase reactant produced mainly by hepatocytes after stimulation by cytokines such as IL-1, IL-6 and TNF-α [64]. CRP down-regulates synthase endothelial nitric oxide (eNOS) transcription in EC, resulting in decreased NO release [80]. Moreover, CRP was found to increase ICAM-1, VCAM-1, and E-selectin expression in HUVEC [81]. Consistently, several clinical trials have reported that CRP levels are associated with endothelial dysfunction and different stages of CAD [64,82]. Higher hsCRP plasma levels were associated with coronary endothelial dysfunction, suggesting it is an independent marker of abnormal coronary vasoreactivity in patients with non-obstructive coronary disease [83]. Recently, high hs-CRP levels were found to correlate positively with IL-6 and LDL-cholesterol and increased CAD risk and mortality [84].

*Fibrinogen*. The canonical role of the hemostatic and fibrinolytic systems is to maintain vascular integrity. Perturbations in either system can prompt primary pathological end points of hemorrhage or thrombosis with vessel occlusion. However, fibrinogen and proteases controlling its deposition and clearance, including prothrombin and plasminogen, have powerful roles in driving acute and reparative inflammatory pathways that affect the spectrum of tissue injury, remodeling, and repair. Although thrombin can clearly influence inflammatory events through protease-activated receptors signaling pathways, fibrinogen as a downstream target of thrombin is one of the most potent contributors among all coagulation system proteins to the inflammatory response [85]. Fibrinogen is a circulating glycoprotein that plays a role in wound healing, thrombosis, platelet aggregation, and inflammation, as well as cell adhesion, vasoconstriction, and chemotactic activity [86]. Increased plasma fibrinogen levels are associated with an increased risk of CAD [87] and myocardial infarction [88]. The correlation between fibrinogen levels and CAD risk has been consistently documented by several prospective studies that include hundreds of cases and controls [89,90]. In addition, recent studies have reported that fibrinogen plasma concentrations show a positive correlation with the degree and composition of the coronary plaque [91,92] but not with atherosclerotic plaque vulnerability as diagnosed by optical coherence tomography [93]. Moreover, elevated fibrinogen levels were associated with CAD severity and mortality [92,94,95], and are a well-established independent risk factor for CAD. Interestingly, epidemiological studies have specifically associated increased circulating levels of the fibrinogen γA/γ′ isoform with arterial and venous thrombosis risk and with increased incidence of coronary artery disease [96], myocardial infarction [97] and ischemic stroke [98], suggesting that γA/γ′ fibrinogen promotes arterial thrombosis. Nevertheless, a prospective study found that alternatively spliced γ′ fibrinogen does not influence CVD events through its prothrombotic properties, and that γ′ fibrinogen concentrations seem to reflect general inflammation that accompanies and may contribute to atherosclerotic CVD [99]. Future studies are required to continue exploring the value of fibrinogen as a biomarker for inflammation and/or thrombosis and cardiovascular disease.

*Serum Amyloid A (SSA)*. Serum amyloid A (SAA) refers to a family of proteins encoded by several genes. Humans have 4 SAA genes (SAA1, SAA2, SAA3, and SAA4), and only SAA1 and SAA2 encode acute phase proteins that are highly inducible during the acute-phase response. These proteins have pro-inflammatory and pro-atherogenic properties and are well known for being involved in atherosclerosis development [100]. Several clinical studies have evaluated the role of SAA levels in CAD severity and future cardiovascular events [101,102,103]. In the Women’s Ischemia Syndrome Evaluation study, elevated circulating SAA levels were associated with angiographic severity of CAD and were highly predictive of a 3-year risk for cardiovascular events [102]. The Thrombogenic Factors and Recurrent Coronary Events (THROMBO) study reported that SAA levels measured two months after myocardial infarction, were associated with infarction severity, with only a trend for increasing the risk of recurrent coronary events over two years [101]. Moreover, a prospective cohort study reported that serum levels of the SAA/LDL complex were associated with an increased risk of a future cardiac event in patients with stable CAD, suggesting SAA/LDL complex could be a more sensitive marker than CRP or SAA for the prediction of prognosis in CAD patients [103]. More recently, elevated SAA1 levels were associated with the presence of acute coronary syndrome (ACS) and with the severity of CAD in ACS patients [104], and high SAA plasma levels were found to be associated with unstable CAD in the prospective Ludwigshafen Risk and Cardiovascular Health (LURIC) study [105]. Altogether, these studies underline the importance of inflammation in CAD and the prognostic relevance of inflammatory biomarkers in CAD severity and recurrent cardiovascular events.

#### 3.1.2. Cytokines

Cytokines include many pleiotropic proteins that have been extensively implicated in the process of inflammation and atherosclerosis. Pro-inflammatory cytokines such as TNF-α, IL-6, IL-8, and IL-18 are essential inflammatory mediators, aggravating the inflammatory response and inducing the expression of adhesion molecules in endothelial cells and leukocytes (E-selectin, and P-selectin), which worsen endothelial dysfunction.

*Tumor Necrosis Factor alpha (TNF-α)*. TNF-α plays an important role in endothelial dysfunction and inflammation [106]. TNF-α activates endothelial cells to express and release various inflammatory cytokines, chemokines and adhesion molecules necessary for the recruitment and migration of monocytes to the vessel intima [107]. In clinical studies, increased TNF-α circulating levels were found to be associated with the endothelial dysfunction in hypertensive patients [108]; with carotid atherosclerosis thickness in patients with early atherosclerosis [109] and with CAD risk [110,111]. Moreover, increased plasma TNF-α levels were found to activate monocytes and to directly trigger foam cell formation [112].

*Interleukin 6 (IL-6)*. Interleukin-6 is an important cytokine involved in many different immunological processes such as the major regulator of acute phase response proteins and plays a crucial role in CAD [113,114]. Although there are few studies of IL-6 as a biomarker in CAD, it seems to have a potential role as a biomarker in coronary plaque instability and has been associated with clinical outcomes in CAD patients. In individuals with ischemic heart disease, IL-6 plasma levels were associated with the presence of thin-cap fibroatheroma observed with optical coherence tomography, and IL-6 levels showed higher sensitivity and specificity than hs-CRP for the prediction of plaque instability [115]. In addition, in a sub-analysis of the STABILITY (stabilization of atherosclerotic plaque by initiation of darapladib therapy) trial, plasma IL-6 levels in patients with stable CAD were associated with several health outcomes, including cardiovascular and all-cause mortality, myocardial infarction and heart failure [116]. Future studies are required to further explore and validate the value of IL-6 levels as a biomarker for endothelial dysfunction and cardiovascular disease.

*Interleukin 8 (IL-8)*. IL-8 is a potent chemoattractant for neutrophils and T lymphocytes. Several studies showed that elevated IL-8 levels may be useful as a biomarker for CAD [117,118,119]. Interestingly, a prospective study including angiographically confirmed stable CAD patients reported that among ten cytokines (IL-1β, IL-2, IL-4, IL-5, IL-6, IL-8, IL-10, TNF-α, granulocyte-macrophage colony stimulating factor and IFN-gamma) and hs-CRP, only elevated IL-8 serum levels were considered as a potential marker of long-term outcome [120]. This suggests that IL-8 may be a predictor of cardiovascular events independent of other cytokines and hs-CRP in patients with CAD.

*Interleukin 18 (IL-18)*. IL-18 is a pro-inflammatory cytokine that belongs to the IL-1 superfamily. It is produced mainly by macrophages and has pleiotropic functions inducing the expression of other cytokines implicated in atherosclerosis [121]. In 2001, Mallat et al. for the first time reported significant IL-18 expression in human carotid atherosclerotic plaques and found an association between IL-18 mRNA transcript levels and clinical and pathological signs of plaque instability [122]. Moreover, plasma or serum IL-18 concentrations have been significantly associated with coronary plaque area in postinfarction patients [123], with fatal cardiovascular events over a 4 year follow up in patients with stable and unstable angina [124] and with CAD in a case-control study performed in the Chinese population [125]. Consistently, serum IL-18 levels were found to be a predictor of coronary events in healthy European men [126] and two independent meta-analyses identified an association between IL-18 and CAD, suggesting circulating IL-18 levels as a prospective and independent marker of CAD risk [110,127].

Other circulating cytokines have been recently proposed as promising CAD markers such as IL-37 levels, which were increased in patients with acute coronary syndrome and associated with a worse clinical outcome after ST-segment elevation acute myocardial infarction [128]. Although a wide range of data that support the potential roles of cytokines as CAD biomarkers, further data from clinical and epidemiologic studies are needed to assess the role and predictive values of cytokines or cytokine combinations in CAD.

#### 3.1.3. Cell Adhesion Molecules (CAM)

In EC, inflammatory cytokines induce increased expression of CAM such as E-selectin, P-selectin, ICAM-1, and VCAM-1. These are transmembrane proteins that promote endothelial dysfunction and leukocyte migration [129,130]. Activated EC produce soluble types of these adhesion molecules that are secreted into the bloodstream. Many studies have reported increased circulating levels of soluble E-selectin, VCAM-1 and/or ICAM-1 associated with CAD, CAD severity and complications [129,131,132,133,134,135,136,137,138,139,140]. However, because most of these CAM are produced not only by the endothelium but also by leukocytes and platelets, these molecules are not specific biomarkers accurately reflecting endothelial damage [141,142]. Therefore, although circulating CAM reflect early inflammation and EC activation, they have a limited diagnostic value when measured alone [130].

#### 3.1.4. Cellular Markers

Microparticles (MP) are small vesicles (0.1 to 1 µm) shed from the plasmatic membrane of different cells, such as erythrocytes, leukocytes, platelets, and EC [130]. Increased circulating endothelial MP (EMP) concentrations have been associated with endothelial dysfunction in patients with CAD [143] and with ACS [144]. Moreover, patients with myocardial infarction registered 79% higher production of EMPs compared to patients with stable CAD [145]. Interestingly, EMP concentrations in patients with ACS and stable CAD patients were higher in the coronary than in the peripheral circulation [146]. This is relevant because experimental evidence using EMPs derived from dysfunctional endothelial cells suggests that EMPs might be involved in the regulation of monocyte/macrophage function producing pro-inflammatory cytokines [144].

Monocytes have also been identified as potential markers of vascular dysfunction and CAD [147,148,149]. In patients with CAD, higher monocyte counts were associated with increased risk of cardiovascular events and with peripheral endothelial dysfunction measured by peripheral arterial tonometry [147]. Regarding specific monocyte subsets, in a group of CAD patients undergoing coronary artery bypass grafting, endothelial dysfunction was associated with higher expression of activation marker CD11c selectively on CD14+CD16++ monocytes, which were associated with more advanced vascular dysfunction measured as NO-bioavailability and vascular ROS production [148]. Moreover, in patients undergoing cardiac catheterization, CAD severity was directly associated with CD16+ monocytes and inversely associated with M2 macrophages, suggesting that measures of monocyte and macrophage subtypes could also be potential biomarkers in CAD [149].

#### 3.1.5. Others

*Asymmetric dimethylarginine (ADMA) and Symmetric dimethylarginine (SDMA)*. ADMA is an endogenous inhibitor of NOS via competition with L-arginine, which is a NOS substrate and a structural analogue of ADMA [150]. Diminished nitric oxide synthesis due to eNOS downregulation appears to play a prominent role in vasoactive factor imbalance, impairment of endothelial hemostasis and the early development of atherosclerosis [151]. ADMA is one of the most potent endogenous inhibitors of the three isoforms of NOS and is involved in the pathogenesis of an array of diseases including systemic sclerosis and rheumatoid arthritis, where derangement of nitric oxide/ADMA pathway has been observed [152,153]. ADMA has been recognized as a hallmark of endothelial dysfunction and related to many cardiovascular risk factors such as hypertension [154], obesity [155], hypertriglyceridemia [156] and type 1 and type 2 diabetes mellitus (T2DM) [157,158]. Furthermore, longitudinal studies have revealed that ADMA predicts cardiovascular morbidity and mortality in both myocardial infarction [159] and stroke patients [160]. Elevated plasma ADMA concentrations not only correlate with the presence of plaques, particularly in the carotid artery, but also predict the risk of future lesion development, myocardial infarction and stroke [161]. Moreover, serum ADMA levels significantly correlated with the presence and extent of coronary atherosclerosis in patients undergoing elective coronary angiogram. Patients with reduced endothelial function in at least one vessel as measured by fractional flow reserve had significantly higher ADMA levels than patients without functionally significant CAD, concluding that serum ADMA levels are independent predictors of coronary atherosclerosis extent and functional significance [162].

Even though SDMA is not a direct inhibitor of NOS, it can alter L-arginine concentrations via competition with amino acid transporters [161]. SDMA was initially thought to be biologically inert and was therefore included in early epidemiological studies as a negative control [163]. Afterwards, SDMA was found to be a sensitive parameter of renal function, sometimes even more sensitive than creatinine [164]. Interestingly, Schulze et al. showed that SDMA predicts all-cause mortality following ischemic stroke even after adjustment for renal function and novel risk factors such as C-reactive protein, albumin, beta-thromboglobulin, and the von Willebrand factor [165]. It has also been proposed that SDMA accumulates in high-density lipoprotein (HDL) fractions from patients with chronic kidney disease activating the toll like receptor 2 (TLR-2) and enhancing an endothelial proinflammatory response [166]. Finally, in a prospective observational study, homoarginine (a homologue of L-arginine)/ADMA ratio and homoarginine/SDMA ratio were inversely associated with cardiovascular death and independent predictors of cardiovascular mortality in claudicant patients with lower extremity arterial disease. Further investigations are needed to explore whether the homoarginine/SDMA ratio is an appropriate predictor for cardiovascular death in patients with other cardiovascular diseases [167].

### 3.2. Novel Potential Biomarkers

Systemic biomarkers are invaluable for basic and clinical research. Numerous studies have assessed the role of cells, proteins, lipids and other metabolites in the pathophysiology of endothelial dysfunction and atherosclerosis, as well as their potential usefulness in the clinical setting. Recently, liquid biopsies and new molecular biology techniques have identified many novel potential biomarkers of inflammation and endothelial dysfunction biomarkers associated with CAD, described in Table 1 [140,160,161].

The growing list of potential biomarkers associated with CAD, ACS, STEMI and other cardiovascular phenotypes are promising, but require validation in future studies.

### 3.3. Therapeutic Strategies by Prevent Endothelial Dysfunction

Maintaining endothelial function is essential to prevent CAD and promote cardiovascular and cardiometabolic health. Some studies suggest that endothelial dysfunction may be prevented and even reversed by drug treatment, alternative therapies and lifestyle changes [199].

#### 3.3.1. Pharmacological Therapy

Endothelial dysfunction is a process that precedes atherosclerosis. In this context, it is essential to understand the mechanism of drugs to prevent endothelial dysfunction. Medical treatments can have short and long-term effects, and some have pleiotropic effects. These effects may act directly on EC and reduce cardiovascular risk. There are many risk factors that can accelerate development of CAD such as hypertension, hyperlipidemia and hyperglycemia which are modifiable by medical treatment [61].

*Glucose lowering drugs*. Several glucose lowering drugs have been tested for prevention of endothelial dysfunction both in vitro and in vivo. In human abdominal aortic endothelial cells, the effect of empagliflozin, a sodium-glucose co-transporter-2 inhibitor (SGLT2i), preserved glycocalyx integrity, increased heparan sulphate synthesis and restored the mechanotransduction response of ECs with damaged glycocalyx. Moreover, empagliflozin promoted an anti-inflammatory phenotype of these cells under pro-inflammatory conditions [200]. On the other hand, in nondiabetic apolipoprotein E deficient (*Apoe-/-*) mice, the dipeptidyl peptidase-4 (DPP-4) inhibitor vildagliptin reduced VCAM-1 expression and attenuated atherosclerotic lesion progression in the aortic arch, and improved EC function by the activation of eNOS [201]. Metformin is a biguanide broadly used for the treatment of T2DM. The effect of metformin therapy on endothelial function was evaluated in prediabetic patients with stable angina and nonobstructive coronary stenosis and was found to decrease blood concentrations of inflammation markers such as IL-1, IL-6, TNF-α and CRP in these patients [61]. In a prospective study, metformin treatment reduced the frequency of major adverse cardiac events after 24 months of follow-up in prediabetic patients [202]. Therefore, the use of glucose-lowering drugs in early stages are likely to prevent endothelial dysfunction and progression of atherosclerosis.

*Statins*. Statin therapy reduces endogenous cholesterol synthesis by inhibiting hydroxymethylglutaryl coenzyme A reductase activity. Simvastatin is a commonly used hypolipidemic drug with effects on EC function. A low simvastatin dose increased HUVEC viability and reduced VCAM-1 and ICAM-1 expression, while statin therapy inhibited endothelial reticulum stress by reducing intracellular cholesterol accumulation and blocking intracellular signal transduction in the Wnt/β-catenin pathway in these cells [203]. Furthermore, atorvastatin and curcumin co-delivered by liposomes decreased E-selectin and ICAM-1 expression in dysfunctional human aortic endothelial cells (HAEC). When administered to *Apoe-/-* mice, this combination therapy reduced plasma lipid levels and reduced IL-6 expression in aortic artery samples [204]. Thus, several lines of evidence show that statins may prevent endothelial dysfunction by decreasing the expression of adhesion and inflammation molecules, although clinical studies are necessary to validate these results.

*Anti-inflammatory drugs*. Because atherosclerosis is a chronic inflammatory disease, novel anti-inflammatory drugs may be useful to prevent endothelial dysfunction and CAD. Tocilizumab is a monoclonal human antibody that blocks IL-6 receptors and is used as treatment for rheumatoid arthritis (RA). Bacchiega and colleagues completed a clinical trial in 18 patients with RA who received tocilizumab therapy for 4 weeks and found decreased CRP levels in blood [205]. Moreover, in an independent study with RA patients, tocilizumab treatment was found to increase endothelial glycocalyx thickness as measured by perfused boundary region and to reduce arterial stiffness [206]. However, although tocilizumab treatment improved markers of endothelial dysfunction, both studies reported increased lipid levels, a well-known risk factor for CAD.

The Janus kinase inhibitor tofacitinib was found to decrease VCAM-1, ICAM-1, TNF-α and IL-1β expression and increase cellular viability of oxLDL-stimulated HAEC [207]. Furthermore, the tyrosine kinase inhibitor imatinib was found to ameliorate endothelial dysfunction in rabbits fed a high cholesterol diet. In this model, imatinib treatment improved acetylcholine-induced relaxation of isolated rabbit aortic rings and increased aortic NO content. In addition, imatinib treatment decreased blood CRP and lipid levels [208]. Finally, zafirlukast is a potent anti-inflammatory mediator that blocks the type 1 cysteinyl leukotriene receptor (CysLT1R). Zafirlukast therapy was found to decrease ICAM-1, VCAM-1, IL-1, IL-6, and IL-8 expression, and to suppress TNF-α-induced production of ROS in HAEC cells treated with TNF-α, suggesting that zafirlukast treatment may eventually be used to decrease cardiovascular risk [209].

*Antiplatelet drugs*. Evidence from various studies show that antiplatelet drugs may prevent endothelial dysfunction by several mechanisms. Vorapaxar is an antagonist and protease-activated receptor 1 (PAR-1) inhibitor. In EC stimulated with cholesterol, PAR-1 inhibition significantly increased EC viability; decreased IL-1β, IL-8 and TNF-α levels and increased IL-13 (anti-inflammatory interleukin) expression. Moreover, vorapaxar decreased DNA damage and increased eNOS expression via the AKT/JNK signaling pathway in these cells [210]. Ticagrelor is a P2Y12 inhibitor. In a study of patients with stable CAD and chronic obstructive pulmonary disease, ticagrelor combined with aspirin was found to better improve endothelial function parameters (apoptosis rate and NO levels in HUVEC treated with patient’s serum, and intracellular ROS levels in peripheral blood mononuclear cells isolated from patients) as compared to clopidogrel and aspirin combined therapy [211]. Moreover, ticagrelor treatment decreased the circulating levels of epidermal growth factor (EGF), a marker of endothelial dysfunction. Serum of ticagrelor-treated patients was found to increase eNOS levels of HUVEC [212]. Shortly afterwards, peripheral blood cells from CAD patients with concomitant chronic obstructive pulmonary disease treated with ticagrelor showed decreased SIRT1 and HEAS1 mRNA expression (anti-inflammatory genes), which correlated negatively with EGF serum levels [213].

*Others*. The effects of the selective serotonin agonist (R)-2,5-dimethoxy-4-iodoamphetamine ((R)-DOI) were assessed in *Apoe-/-* mice fed with high fat diet. Sustained delivery of low-dose (R)-DOI decreased serum cholesterol levels and *vcam-1, IL-6* and *tnf-α* gene expression in aorta arch of these mice, suggesting that (R)-DOI could counteract the pro-inflammatory effects induced by Western diet [214]. On the other hand, fenofibrate is a peroxisome proliferator-activated receptor alpha agonist widely used to treat mixed dyslipidemia. In mice with induced diabetes, fenofibrate treatment significantly ameliorated acetylcholine-induced endothelium dependent relaxation of mouse aortae. Furthermore, in mouse aortic endothelial cells cultured with high glucose, fenofibrate therapy increased NO production, decreased superoxide anion levels, and activated eNOS and adenosine monophosphate-activated protein kinase (AMPK) phosphorylation, suggesting that fenofibrate improves endothelial function [215].

Altogether, these drugs may be promising therapies to prevent proinflammatory state and endothelial dysfunction and stop development and progression of CAD; however, more clinical trials are required to ensure efficacy and safety.

#### 3.3.2. Non-Pharmacological Therapies

##### Lifestyle Modifications

Lifestyle plays a key role in the development of endothelial dysfunction and related diseases. An illustrative example is the increasing incidence of cardiovascular events and prevalence of overweight and metabolic syndrome (MS) in Japanese society by lifestyle westernization [216]. Given the influence of lifestyle, in addition to pharmacological intervention, conventional treatment of endothelial dysfunction includes lifestyle modification (exercise, diet and smoking). We describe here recent scientific evidence supporting the impact of lifestyle modification in patients with endothelial dysfunction and related disease.

*Exercise*. Currently, exercise is a well-recognized component of the integral therapy of cardiac rehabilitation and an effective element to restore endothelial function. Molecular effects of exercise training in patients with cardiac disease have been thoroughly investigated [217]. The effects of exercise training effects on endothelium function include increased eNOS and NO bioavailability, and reduced ROS [217]. In 2013, the American Heart Association (AHA) updated its exercise training guideline for suitable patients as an adjuvant to cardiac therapy. These standards are meant to be used by physicians, nurses and other health care professionals to treat patients with cardiovascular disease and other disease states [218]. Various exercise programs have improved endothelial function in patients with cardiovascular disease [219,220,221]. In men, regular exercise preserves vascular endothelial function with ageing; but in estrogen-deficient postmenopausal women the evidence is not consistent. However, there is evidence indicating that estradiol treatment restores the ability of exercise training to improve endothelial function in these women [222,223].

*Diet*. Many endothelial dysfunction-related diseases are associated with poor quality diet. These diseases include cardiovascular disease (CVD), T2DM, MS and obesity. Health organizations have been promoting dietary recommendations to reduce CVD risk [224,225,226]. The “2017 AHA presidential advisory on dietary fats and CVD” and the “2015 to 2020 Dietary Guidelines for Americans” draw attention to reducing the intake of low saturated fats and replacing it with unsaturated fats (particularly polyunsaturated fats), adopting a healthy eating pattern as the Mediterranean-style and Dietary Approaches to Stop Hypertension (DASH) [224,226]. The Mediterranean diet has been extensively studied and associated with a lower risk of all-cause mortality, cardiovascular disease, T2DM, and some types of cancer [227,228,229,230,231,232]. This diet is characterized by high consumption of vegetables, legumes, fruits, nuts, whole grains, fish, seafood, extra virgin olive oil, a moderate intake of red wine, low consumption of red and processed meat; as well as low consumption of sugar (sucrose), sweetened foods and drinks and refined grains [233]. Bioactive components of the Mediterranean diet include fiber, phytosterols, polyphenols, monounsaturated and polyunsaturated fats, vitamins and minerals [234], which provide its lipid-lowering, insulin-sensitizing, antioxidative, anti-inflammatory and antithrombotic properties [233].

*Smoking*. According to the World Health Organization global report on trends in prevalence of tobacco use 2000–2025 third edition, around 8 million people died from tobacco-related disease in 2017, and even though the smoking population is decreasing in developed countries, it is growing in developing countries [235]. Smoking is one of the major risk factors of cardiovascular diseases, and oxidative stress is the most likely underlying mechanism of smoking-induced endothelial dysfunction [236]. Even low cigarette consumption (1 cigarette per day) represents a risk factor for developing coronary heart disease and stroke [237]. While the use of electronic cigarettes was initially thought to be safe, a clinical study revealed that biomarkers for oxidative stress and flow-mediated dilatation were equally affected in both traditional and electronic cigarette smokers [238]. Moreover, the results of a systematic review and meta-analysis suggested that the electronic cigarettes should not be labelled as cardiovascular safe products [239]. However, nicotine replacement therapy can be of aid to restore endothelial function after smoking cessation [240].

##### Antioxidant Therapy

Endothelial dysfunction is a multifactorial disease, where ROS production is one of the mechanisms involved. ROS are synthetized in the endothelial cell layer, and within the media and adventitia. The imbalance of ROS synthesis and the antioxidant mechanisms in favor of ROS formation is the definition of oxidative stress [241]. Oxygen toxicity has an impact on all major organic molecules such as nucleic acids, proteins and free amino acids, lipids and carbohydrates. Molecular mechanisms involved in endothelial dysfunction by oxidative stress include the uncoupling of eNOS by reactive oxygen and nitrogen species, upregulation of ET-1 expression with subsequent production superoxide/hydrogen peroxide and activation of NADPH oxidase (Nox1 or Nox2) via angiotensin II [242,243,244]. Historically, antioxidative vitamin (C and E) administration has contributed to restore endothelial function and thus to decrease the risk of cardiovascular disease [242]. For example, high dose intravenous vitamin C restored blood flow during endotoxemia in healthy human subjects, presumably preventing NO inactivation by ROS [245]. On the other hand, endothelial dysfunction is characterized by decreased NO bioavailability, most likely originated by increased arginase activity. Several studies have focused on the effects of plant-derived compounds such as polyphenols (known to inhibit arginase activity), for the treatment of different endothelial dysfunction-induced diseases, mainly in animal models. Although clinical trials of these compounds are limited, they have shown promising results.

*Animal models*. Fauzy et al. [246] observed that treatment of hypertensive rats with an aqueous extract of the plant *Piper sarmentosum*, significantly reduced blood pressure. Mesenteric artery NO levels were increased and vasoconstrictor ET-1 were reduced after 28 days of treatment. Chemical constituents of *Piper sarmentosum* include quercetin, a flavonoid that modulates the ET-1 and NO circulating levels, presumably via NADPH oxidase inhibition and eNOS activation. Furthermore, ginger is well known for its beneficial properties preventing cardiovascular disease. In order to identify the possible molecular mechanism underlying these properties, effects of aqueous extracts from two different ginger varieties (*Zingiber officinale* and *Curcuma longa*) were assessed in a high-cholesterol-diet hypercholesterolemic animal model, characterized by increased arginase activity, atherogenic index and plasma lipid levels. Composition analysis of these extracts revealed the presence of several antioxidant compounds, including high concentrations of polyphenolic compounds. Rats with an extract supplemented-diet showed decreased plasma and liver arginase activity, atherogenic indices and plasma lipid levels. Authors proposed the mechanism was mediated by the antioxidant effects of polyphenols [247]. Likewise, 2,3,5,4′-tetrahydroxystilbene-2-O-β-D-glucoside (THSG), a compound extracted from the rhizome of *Polygonum multiflorum* with dose-dependent arginase inhibitory activity, showed increased NO synthesis and reduced ROS levels in the aortic endothelium of both wild-type and atherosclerosis-prone *Apoe-/-* mice [248].

*Studies in Humans*. *Phyllanthus emblica* fruit aqueous extracts were tested in a clinical trial of subjects with MS. Subjects who received the extract showed significant improved levels of endothelial dysfunction, oxidative stress and systemic inflammation, as well as lipid profiles. Previous reports showed that these extracts inhibit the production of lipid peroxide and is a scavenger of hydroxyl and superoxide radicals in vitro and in vivo [249]. Furthermore, these extracts improved endothelial function and reduced oxidative stress and systemic inflammation biomarkers in patients with T2DM [250,251].

In a randomized crossover clinical trial, pre-hypertensive patients (systolic blood pressure between 125- and 160-mm Hg) consumed pure flavonoids (a subclass of polyphenols) epicatechin and quercetin and serum biomarkers of endothelial dysfunction and inflammation were assessed at the end of the study. Epicatechin improved endothelial dysfunction with no effect on inflammation, whereas quercetin reversed endothelial dysfunction and reduced inflammation [252]. Another randomized, double-blind, cross-over, placebo-controlled clinical trial in hypertensive patients, showed gender differences in endothelial dysfunction by resveratrol intake. Only women presented a significant improvement in endothelial function, with no changes in blood pressure measurements in either gender. The authors suggested that this gender difference could be explained by lower levels of oxidative stress in women [253].

The goal of these studies was to find alternative treatments to restore endothelial function with minimal adverse effects. Since these plant-derived compounds have been used in traditional medicine as herbal therapy to treat several conditions including those related to endothelial dysfunction, they may represent a safe alternative that could be used as adjuvant or substitute of conventional drugs.

## 4. Conclusions

CAD is the most common form of heart disease. It is the result of atheromatous changes in the vessels supplying blood to the heart. CAD describes a range of clinical disorders from asymptomatic atherosclerosis and stable angina to acute coronary syndromes. The endothelium plays an important role in maintaining vascular homeostasis and regulating blood vessel function. Evidence supports the central role of endothelium and inflammation in all phases of the atherosclerotic process. It is now clear that inflammatory processes have a key role not only in the initiation and progression of atherosclerosis but also in the stability of the established atherosclerotic plaques. Early atherosclerotic lesions are reversible, and the use of liquid biopsies and new molecular biology techniques have allowed the identification of molecular and cellular markers of endothelial dysfunction such as EMPs, microRNAs, phospholipids and proteins that help establish an early diagnosis and may be targets to prevent endothelial dysfunction and CAD. Additionally, pharmacological and non-pharmacological therapeutical approaches can improve endothelial function, prevent atherosclerosis and CAD, and avoid its complications.

## Figures and Tables

**Figure 1 ijms-22-03850-f001:**
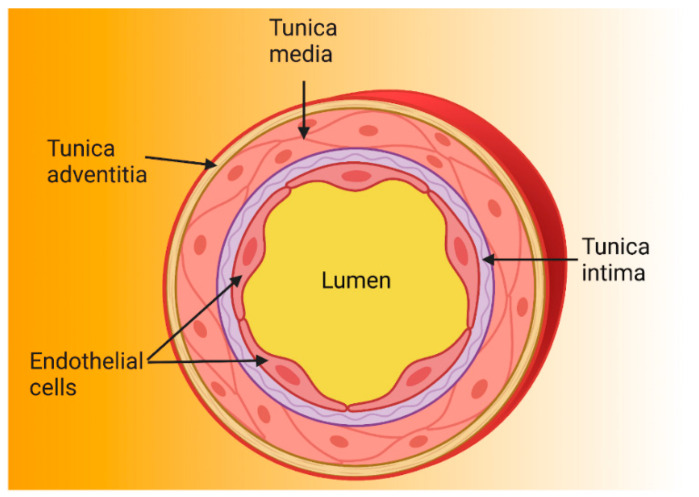
Structure of the arterial wall. Because arteries are supplying blood to tissues and are exposed to high pressure, their walls are thicker than those of other blood vessels. Arterial walls are composed of three layers: the tunica intima is the innermost layer and is made up of endothelial cells anchored to the basal lamina (connective tissue); the tunica media contains vascular smooth muscle cells and regulates vascular tone; and the tunica adventitia is the outermost and contains nerve endings, perivascular adipose tissue, and connective tissue [27,28,29].

**Figure 2 ijms-22-03850-f002:**
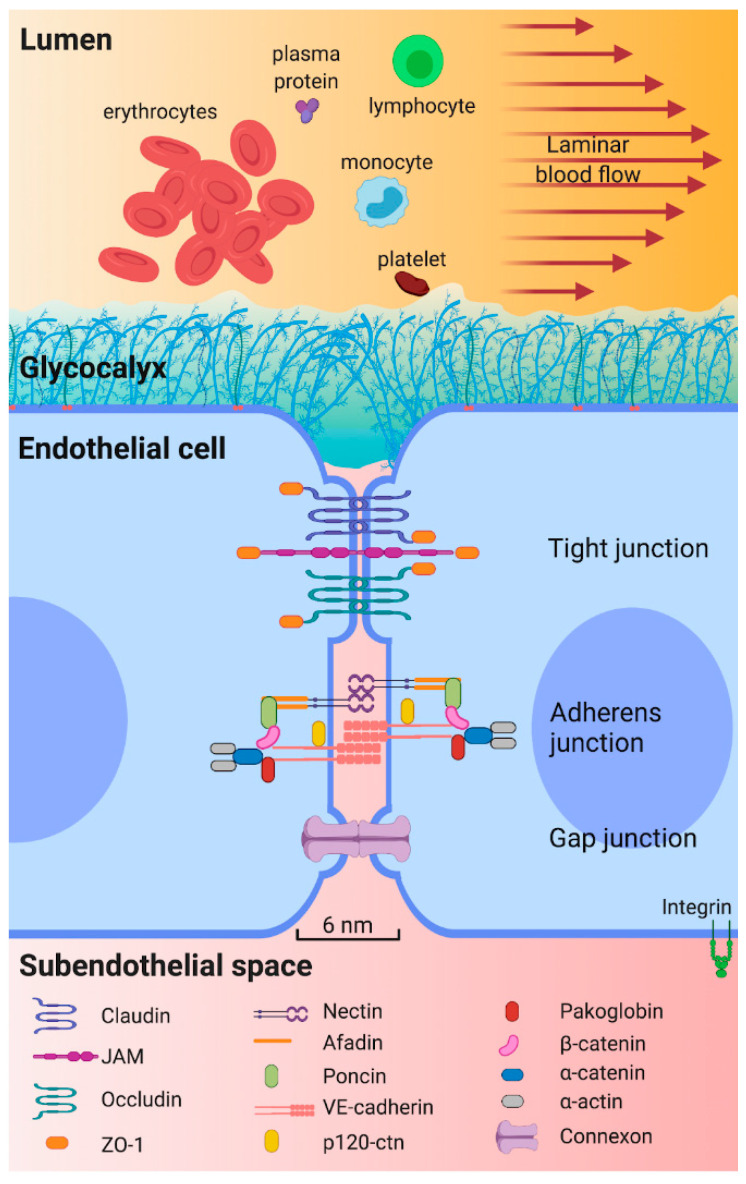
Key structures for epithelium permeability. The arterial endothelium is a semipermeable barrier that prevents passage of blood cells and large molecules from circulating blood to subendothelial space. The glycocalyx and junction complexes are the main structures that help to maintain this function. The glycocalyx, covering the luminal surface of endothelium, comprises proteoglycans, glycosaminoglycans and glycoproteins. Proteoglycans are syndecanes 1, 2 and 4, glypican and perlecan; the main glycosaminoglycans are heparan sulfate, chondritin sulfate, and hyaluronic acid; glycoproteins, involve three families of adhesion molecules, selectin family, the integrin family, and the immunoglobulin superfamily, and their expression depends on the surrounding microenvironment. There are three subtypes of junction complexes connecting adjacent EC. Tight junctions are formed by claudins, junction adhesion molecules (JAM), and occludins. Tight junctions bind to zonule occludens-1 (ZO-1), allowing interaction with cytoskeleton components. Second, adherens junctions are formed by nectin, poncin, afadin, and vascular endothelial cadherin (VE-cadherin) complexes. VE-cadherin binds to filamentous actin (F-actin) via intracellular proteins such as pacoglobin, β-catenin, α-catenin, and α-actin. Third, gap junctions consist of two connexons that form an intercellular channel that physically communicates adjacent endothelial cells (EC) and allows passive diffusion of ions and small molecules [30,40].

**Figure 3 ijms-22-03850-f003:**
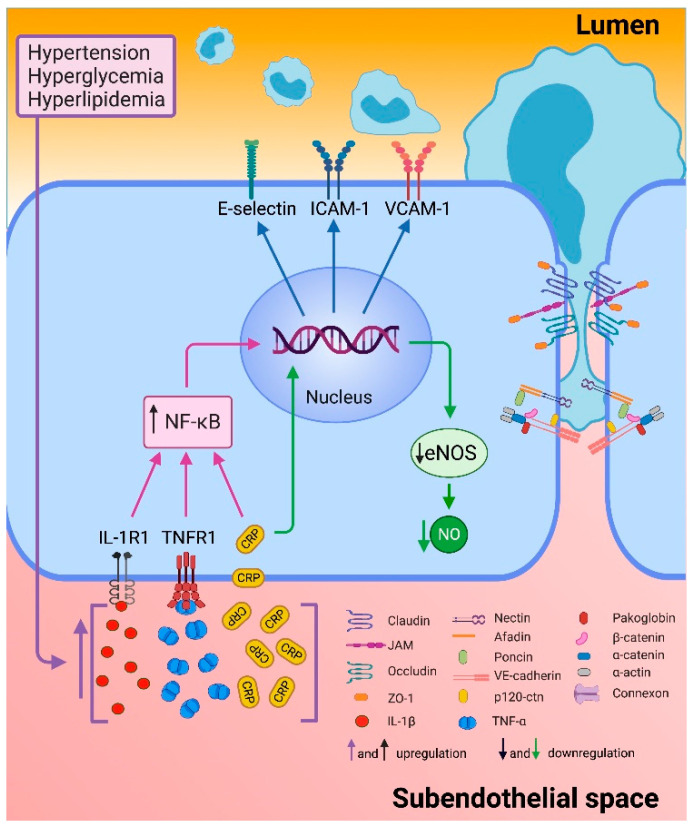
Endothelial inflammation. Endothelial dysfunction is triggered by different cardiovascular risk factors such as hypertension, hyperglycemia, and hyperlipidemia. These events increased production of interleukin 1 beta (IL-1β), tumor necrosis factor alpha (TNF-α), and C reactive protein (CRP). Proinflammatory cytokines bind to their receptors and culminate in the activation of the nuclear transcription factor κB (NF-κB) that stimulate the transcription of selectin-E, intercellular adhesion molecule-1 (ICAM-1) and vascular cell adhesion molecule-1 (VCAM-1). CRP down-regulates endothelial nitric oxide synthase (eNOS) transcription and destabilizes eNOS mRNA, resulting in decreased nitric oxide (NO). Furthermore, the reorganization of actin filaments allows the opening of intercellular junctions through other signaling pathways [63,64].

**Table 1 ijms-22-03850-t001:** Recently identified potential biomarkers of inflammation and endothelial dysfunction in coronary artery disease (CAD).

Biomarker	Model	Implication or Considerations	Study
Adiponectin	Cohort of T1DM patients	↑Adiponectin serum levels in T1DM patients are proposed as an early marker of subclinical atherosclerosis.	[168]
ANGPTL8	Cohort of CAD patients	CAD patients had significantly higher serum ANGPTL8 levels and ANGPTL8 was independently associated with TG and ICAM-1 in CAD patients.	[169]
CTRP9	Cohort of patients with CAD and T2DM	↑expression of CTRP9 in patients with CAD and T2DM.	[170]
Cyr61	Cohort of CAD patients	Serum Cyr61 levels were higher in CAD patients than in controls and correlated positively with Gensini score and CRP levels.	[171]
Endocan	Cohort of patients with isolated CAE Cohort of patients with CSX	Plasma endocan levels were increased in patients with isolated CAE as compared to controls. Endocan serum levels were higher in CSX patients than in controls and are proposed as an acute marker of microvascular disease.	[172] [173]
Galectin-3	Cohort of CAD patients	Serum galectin-3 levels were higher in CAD patients than controls and were associated with severity of CAD.	[174]
Human neutrophil peptides or α-defensin	Cohort of CAD patients, hyperlipidemic patients and controls	Patients with hyperlipidemia and CAD have showed increased α-defensin in blood. α-defensin is proposed as a potential inflammation marker that may predict the risk of CAD.	[175]
Irisin	Cohort of obese children.	Obese children showed a decreased level of Irisin as compared to lean children. ↓Irisin levels correlated inversely with several markers of inflammation and endothelial dysfunction in obese children.	[176] [177] [178]
	Cohort of children and adolescents with T2DM, MS and controls	T2DM and MS patients showed decreased levels of Irisin as compared to healthy controls. Irisin levels showed a negative correlation with sVCAM-1, sICAM-2 and MCP-1 in the total population of children and adolescents.	
	Meta-analysis of 7 case-control studies involving 867 patients of CAD and 700 controls	Circulating irisin concentrations were 18.10 ng/mL lower in patients with CAD than in healthy controls.	
Lp-PLA2	Cohort of patients with CAD + ASC	Circulating plasma Lp-PLA2 levels were higher CAD+ACS patients than in controls and showed a positive association with CAD risk.	[179]
NGAL and YKL-40	Prospective study of T2DM patients.	NGAL and YKL-40 serum levels are increased in T2DM subjects with subclinical CAD and are associated with the risk of future cardiovascular events.	[180]
Resistin and Visfatin	Cohort of CAD patients.	Resistin and visfatin serum levels were higher in patients with acute myocardial infarction than in patients with stable angina.	[181] [182]
	Cohort of patients with CAE.	Visfatin serum levels were higher in patients with both CAE + CAD and are proposed as an independent marker for severity of coronary ectasia in both isolated CAE and CAD coexisting with CAE groups.	
Renalase	Patients presenting to the emergency room with acute chest pain, with diagnostic workup including PET to identify CMD.	↑Renalase serum levels were associated with symptomatic CMD in patients presenting with acute chest pain, increased peripheral renalase blood levels are proposed as a biomarker for CMD.	[183]
Sortilin	Cohort of CAD patients	Sortilin serum levels were higher in CAD patients than in controls and correlated with inflammatory cytokine levels.	[184]
suPAR	Cohort of patients with non-obstructive CAD.	In patients with non-obstructive CAD, plasma suPAR levels correlated negatively with coronary flow reserve. suPAR levels are proposed as an independent risk predictor of coronary microvascular function.	[185]
PCSK9	Cohort of HIV+ patients under retroviral therapy	PCSK9 serum levels were higher in HIV+ than in age and LDL-C level matched HIV-patients and were inversely associated with coronary endothelial function measured by magnetic resonance.	[186] [187]
	Cohort of patients with suspected CAD	Low PCSK9 plasma levels were associated with unfavorable metabolic profile and with diffuse non-obstructive coronary atherosclerosis as determinated by coronary computed tomography angiography.	
Phosphatidylcholine and lysophosphatidylcholine	Cohort of patients with CAD and PAD	Serum of phosphatidylcholine and lysophosphatidylcholine levels were lower in CAD and PAD patients than in controls.	[188]
lncRNA KCNQ1OT1, HIF1A-AS2 and APOA1-AS	Cohort of patients with CAD	KCNQ1OT1, HIF1A-AS2 and APOA1-AS in patients with CAD. ROC analysis confirmed their suitability as biomarkers of CAD.	[189]
Circulating lncRNA IFNGAS1	Cohort of patients with CAD	Increased lncRNA IFNGAS1 plasma levels were associated with CAD risk and severity assessed by coronary angiography.	[190]
Circulating microRNA-941	Cohort of patients with ACS.	microRNA-941 plasma levels were higher in patients with ACS and ST-elevation myocardial infarction than in controls.	[191]
Circulating microRNA-33	Cohort of patients with CAD.	microRNA-33 expression is higher in CAD patients than controls.	[192]
Circulating microRNA-92a	Cohort of patients with T2DM + CAD	↑ Expression of microRNA-92a, was significantly associated with T risk of acute coronary T2DM. miR-92a levels were identified as an independent predictive factor for ACS events in the patients with T2DM.	[193]
Circulating microRNAs-331, 151-3p	Cohort of patients with STEMI.	MicroRNAs-331 and 151-3p were significantly up-regulated in patients with STEMI as compared to patients with stable angina and controls. These miRNAs are proposed as suitable biomarkers than may be associated with plaque rupture.	[194]
Serum exosomal microRNA-21, 126 and PTEN.	Cohort of patients with ACS.	Serum levels of exosomal microRNAs-21, 126 and PTEN were higher in patients with ACS than in controls. Exosomal microRNA-126 showed a positive correlation with coronary artery stenosis severity.	[195]
Circulating microRNA-145	Cohort of patients with ACS.	↓ Expression of microRNA-145 in ACS patients as compared to controls. microRNA-145 levels correlated with other endothelial inflammation and damage markers.	[196]
	ACS coronary ligation rat model	microRNA-145 overexpression in an ACS rat model improved endothelial injury and abnormal inflammation, suggesting it may be a therapeutic target.	
Circulating microRNA-22	Cohort of patients with CSF.	microRNA-22 expression was increased in patients with CSF as compared to those with normal coronary flow. Increased microRNA-22 circulating levels are proposed as a suitable biomarker of CSF.	[197]
microRNA signature	Cohort of patients with CSF.	Expression levels of miR-1, miR-133, miR-208a, miR-206, miR-17, miR-29, miR-223, miR-326, and 155 in PBMCs were significantly increased in SCF patients as compared to controls.	[198]
		Expression levels of microRNAs: miR-15a, miR-21, miR-25, miR-126, miR-16, and miR-18a were significantly decreased in patients with SCF patients as compared to control group.	

Abbreviations: ↑ (increased), ↓ (decreased), T1DM (type 1 diabetes mellitus), ANGPTL8 (angiopoietin like 8), CAD (coronary artery disease), TG (triglycerides), CTRP9 (C1 q/TNF-related protein 9), T2DM (type 2 diabetes mellitus), Cyr61(cysteine-rich protein 61), CRP (C reactive protein), CAE (coronary artery ectasia), CSX (cardiac syndrome X), MS (metabolic syndrome), sVCAM-1 (soluble vascular cell adhesion molecule-1), sICAM-2 (soluble intercellular adhesion molecule 1), MCP-1 (monocyte chemoattractant protein), Lp-PLA2 (lipoprotein associated phospholipase A2), ACS (acute coronary syndrome), NGAL (neutrophil gelatinase-associated lipocalin), YKL-40 (chitinase-3 like protein 1), CMD (coronary microvascular dysfunction), suPAR (soluble urokinase-type plasminogen activator receptor), PCSK9 (proprotein convertase subtilisin/kexin type 9), HIV (human immunodeficiency virus), LDL-C (low density lipoprotein-cholesterol), PAD (peripheral artery disease), lncRNA (long non-coding RNA), KCNQ1OT1 (KCNQ1 opposite strand/antisense transcript 1), HIF1A-AS2 (HIF1A antisense RNA 2), APOA1-AS (APOA1 antisense RNA), STEMI (ST-segment elevation myocardial infraction), PTEN (phosphatase and tensin homolog), CSF (coronary slow flow).

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
