# Peer review of "Endothelial Dysfunction, Inflammation and Coronary Artery Disease: Potential Biomarkers and Promising Therapeutical Approaches"

_ijms, 2021, doi:10.3390/ijms22083850_

Round 1

Reviewer 1 Report

This is an extensive review describing in great detail pathophysiologic mechanisms linking endothelial dysfunction, inflammation and coronary artery disease. The authors suggest a very broad panel of biomarkers as possible tool for the identification and prognosis of atherosclerosis. The issue of the paper is compatible with the aim of the issue of the journal but a number of changes can considerably improve reading.

I believe that  the first part of the review (endothelial function and regulation) could be shortened to ease authors to focus on the main objective of the review namely the biomarkers

As the authors describe one of the major regulators of endothelia function is NO. Inhibitors of NO synthase which could be actively involved in derangement of vascular hemostasis leading to atherosclerotic plaque formation should be mentioned. In that respect asymmetric and symmetric dimethylargines (ADMA and SDMA respectively) should be discussed as potential biomarkers of atherosclerosis with commentary on their relationship with inflammation as stated in the title of the review (Arthritis Res Ther. 2017 Feb 10;19(1):32. Int J Cardiol. 2020 Jan 15;299:7-11, J Clin Med. 2020 Sep 20;9(9):3026, Int J Mol Sci. 2012 Sep 26;13(10):12315-35. Clin Rheumatol. 2010 Sep;29(9):957-64. Int J Cardiol. 2016 Oct 1;220:629-33)

The authors state fibrinogen in acute phase reactants paragraph, however derangement of fibrinolysis and the association with high inflammatory status should be discussed as a mechanism promoting propensity to thrombosis, one of the major events in coronary artery disease (Eur Heart J 2005, 26:332–342  Clin Rheumatol. 2013 Sep;32(9):1357-64.)

Reviewer 2 Report

Dear Author,

Thank you for submitting your manuscript, which is very interesting topic. I think with some work it will be a valuable contribution to the literature.

It is known that increased oxidative stress leads to endothelial dysfunction, so I think that this fact deserves more attention in your work. There are numerous functional studies (animal and human) that indicate an association between endothelial dysfunction and oxidative stress, which leads to a change in the mechanisms of vascular reactivity. Please pay more attention to this in your manuscript.

Furthermore, pediatric hypertension predisposes children to adult hypertension, and is associated with early markers of cardiovascular disease. Irisin is mentioned in your manuscript as a biomarker associated with endothelial dysfunction in children, but I suggest to pay more attention to this topic.

Sincerely,

Round 2

Reviewer 1 Report

Τhe authors have adequately addressed the majority of the comments and the manuscript have considerably improved, I have only a couple of minor comments

The authors should state in their revision letter the word numbering of their manuscript before and after revisions to declare the reduction of the length.

Surprisingly enough the authors have not cited in fibrinogen section the suggested MONICA Optional Heamostasis Study (Eur Heart J 2005;26:332-342) and the DRAGGO study (Clin. Rheumatol 2013;32(9);1357-64) both of which are highly relevant with the subject of the review.
